# Can you refuse these discounts? An evaluation of the use and price discount impact of price-related promotions among US adult smokers by cigarette manufacturers

Ralph S Caraballo, Xu Wang, Xin Xu

National Center for Chronic Disease Prevention and Health Promotion, Office on Smoking and Health, Centers for Disease Control and Prevention, Atlanta, Georgia, USA

**Correspondence to**
Dr Ralph S Caraballo;
rfc8@cdc.gov

## ABSTRACT

**Objectives:** The raising unit price of cigarette has been shown to be one of the most effective ways of reducing cigarette consumption and increasing rates of successful quitting. However, researchers have shown that price-sensitive smokers have used a variety of strategies to mitigate the effect of the rising price of cigarettes on their smoking habits. In particular, 23–34% of adult smokers in the US use cheaper brands, and 18–55% use coupons or promotions. Little is known about the discount use by type of brands. As such, the main purpose of this analysis is to evaluate the uses and price discount effects of these price-related discounts by manufacturers and major brands.

**Setting:** An analysis based on the cross-sectional 2009–2010 National Adult Tobacco Survey (NATS).

**Participants:** 11 766 current smokers aged 18 or above in the USA.

**Primary outcome measures:** Price-related discount was defined as smokers who used coupons, rebates, buy-one-get-one-free, two-for-one or any other special promotions for their last cigarettes purchase.

**Results:** The use of price-related discounts and associated price impact vary widely by cigarette manufacturer and brand. Approximately one of three Camel, one of four Marlboro and one of eight Newport smokers used price-related discounts on their latest cigarette purchases. The average price reductions of discounts offered by Philip Morris (PM) or R.J. Reynolds (RJR) were around 29 cents per pack while that of Lorillard (Newport only) was 24 cents per pack. Cigarette brands that provided significant per pack price reductions include: PM Marlboro (28 cents), RJR brand Camel (41 cents), Doral (50 cents), Kool (73 cents) and Salem (80 cents), and Lorillard Newport (24 cents).

**Conclusions:** Policies that decrease price-minimisation strategies will benefit public health.

**Strengths and limitations of this study**

- Research has shown that increasing the unit price of cigarettes is among the most effective public health interventions to reduce cigarette consumption, prevent smoking initiation and increase rates of successful quitting. This is the first national study in the USA to evaluate the uses and effects of price-related discounts (coupons, rebates, buy-one-get-one-free, two-for-1 or any other special promotions for the last pack of cigarettes purchased) by US cigarette manufacturers and specific cigarette brands.
- The US national study consisted of an analysis of telephone and cell phone cross-sectional data (2009–2010) of 11 766 current cigarette smokers aged 18 or above.
- The price paid for last pack of cigarettes during the past 30 days was collected. As the purchases were made recently (last pack bought in past 30 days and most smokers are daily smokers), recall bias should not be a major problem in this study.
- The study design is cross-sectional. Therefore, the study findings may be specific only to the period October 2009–June 2010.
- The 2009–2010 NATS does not collect information for all price minimisation strategies, including cigarette purchases from states with lower price.
- Smokers' self-reported use of coupon or other types of price-related discounts in the 2009–2010 NATS only reflects direct-to-consumer discounts from the industry. As a result, the discount measure in the analysis does not include tobacco industry's promotional allowances directly paid to cigarette retailers or wholesalers, as these disaggregated promotional allowances by cigarette manufacturers or by brands have not been disclosed to consumers or to the public.

## INTRODUCTION

Cigarette use is the most preventable cause of death and disease in the USA and presents a significant public health burden.[1] Research has shown that increasing the unit price of cigarettes is among the most effective public health interventions to reduce cigarette consumption, prevent smoking initiation and increase rates of successful quitting.[2–7] In addition, recent evidence shows that the 2009 federal tobacco excise tax increases have been one of the strategies that have substantially reduced the number of cigarette and smokeless tobacco users among US middle school and high school students.[8]

Internal documents from cigarette companies have shown that they are aware of the potential impacts that price increases have on their sales and profits. Cigarette companies have developed a variety of price-reduction marketing efforts to promote cigarette sales, such as multipack discounts, rebates and coupons.[9] According to the most recent cigarette report from the Federal Trade Commission, in addition to giving away 50 million cigarettes for free in 2010, the major cigarette manufacturers spent approximately $8.05 billion marketing their products. More than 80% of the marketing expenditures (6.49 billion) went to price-related discounts and promotional allowances used to reduce the retail price of cigarettes.[10]

These cigarette companies' price-related discounts may diminish the public health benefit associated with increased cigarette prices among some smokers even after federal law has raised the unit price of cigarettes. Several recent studies have shown that a large portion of US adult smokers (18–55%) have taken advantage of these price-related discounts offered by some cigarette companies.[11–16] In addition, evidence from other studies has shown that smokers who used these price-related discounts were less likely to make quit attempts or to successfully quit in the future.[17–20] Although studies have previously investigated demographics and socio-economic characteristics of smokers who used price minimisation strategies, including using coupons or other types of discounts from cigarette companies,[11–15] little is known about how these price-related discounts affect the average price paid per cigarette when factoring in discounts offered by specific cigarette companies or when looking at specific cigarette brands. Cigarette companies may be directly influencing the prices of their products by using these types of marketing strategies.

Using unique data from the 2009 to 2010 National Adult Tobacco Survey (NATS) about cigarette brands and price-related discounts used by adult smokers, we evaluated the uses and price discount effects of these price-minimisation strategies by the cigarette manufacturers and major brands. To the best of our knowledge, this is the first study to provide these estimates from a national representative sample of US adult smokers. The findings of the analysis may help policy and public health stakeholders to further understand the promotion strategies of leading US cigarette companies.

## METHODS

### Data source

The 2009–2010 NATS is a stratified, national, landline and a cell phone survey conducted during October 2009–June 2010. The survey population is a representative sample of non-institutionalised adults aged 18 years or older at state and national levels. The survey was developed by the Office on Smoking and Health at the Centers for Disease Control and Prevention and was designed primarily to assess the prevalence of tobacco use and the factors related to tobacco use among US adults. The survey has 130 questions that provide information about demographics, health status, cigarette smoking behaviours, price minimisation behaviours, cigarette brands preference, the use of other tobacco products and attempts at quitting smoking. The 2009–2010 NATS completed a total of 118 581 interviews, including 110 634 by landline and 7947 by cell phone. As samples used for this analysis contain only de-identified observations, this research did not involve human subjects, as defined by Title 45 Code of Federal Regulations, Part 46, and institutional review board (IRB) approval was not required.

This analysis is restricted to current smokers who reported the cigarette brand name that they smoked most often during the past 30 days (n=16 015). Current smokers were defined as those who reported smoking at least 100 cigarettes in their lifetime and currently smoked every day or on some days (n=16 542). Among them, respondents who failed to report a brand name were excluded (n=523). Owing to the concern of small sample size (n=4), respondents who smoked Forsyth, which is a private brand label, were also excluded from the analysis.

In addition, respondents who failed to provide information on price paid for their latest purchase (n=978), the use of price-minimisation strategies (n=2794), demographic characteristics (age, race/ethnicity, gender, education, marital status or employment status; n=477) or time to first cigarette since waking up, were also excluded. The final sample size is 11 766.

### Measures of brands and companies

In the survey, respondents were asked about the cigarette brand that they used most often during the past 30 days. A total of 17 brand choices were listed. Except Forsyth and the choice of other brands (n=3299), the remaining 15 brand names are categorised as premium brands or generic brands. Premium brand names include Camel, Kool, Marlboro, Newport, Pall Mall, Parliament, Salem, Virginia Slims and Winston, and generic brands include Basic, Doral, GPC, Misty, Sonoma and USA Gold.

To evaluate price-related discounts and promotions used by major companies, three major cigarette companies were identified on the basis of the 15 brand names above. They are Philip Morris (PM), R.J. Reynolds (RJR) and Lorillard. These companies jointly represented approximately 85% of total US cigarette sales in 2010.[21] PM's brand names include Basic, Marlboro, Parliament and Virginia Slims. Camel, Doral, GPC, Kool, Misty, Pall Mall, Salem and Winston are the brands manufactured by RJR. Lorillard has the brand Newport. The remaining brands, including Sonoma, USA Gold and the choice of other brands are included under other cigarette companies.

### Measures of prices and discounts from the industry

The 2009–2010 NATS contains two types of price data. Current smokers who bought cigarettes by packs in their latest purchases were asked to report price paid per pack (after discounts or coupons) in dollars. Those who bought cigarettes by cartons were asked to report price paid per carton. Consequently, price per carton was divided by 10 to obtain a consistent measure of price paid per pack.

In the survey, current smokers were also asked whether they had taken advantage of coupons, rebates, buy-one-get-one-free, two-for-one or any other special promotions for cigarettes during the most recent purchase. These coupons and other discount offers were defined in the analysis as coupons and price-related discounts from the industry. Positive responses to this question were used to estimate the prevalence of usage of price-related discounts and promotions by manufacturers and brand names.

### Statistical analysis

Cigarette prices reported in the 2009–2010 NATS may reflect the price paid by a smoker after using multiple price minimisation strategies. To assess the independent price reduction associated with coupons and other price-related discounts directly from the industry, for each cigarette company or brand, regression analysis with the following specification was used to obtain adjusted average prices per pack:

$$\text{Per pack price paid} = \beta_1 + \beta_2 \text{discounts} + \beta_3 \text{Other PMS}$$
$$+ \beta_4 \text{Ciguse} + \beta_5 \text{Demographics}$$
$$+ \beta_6 \text{state}$$

The dependent variable is per pack price paid for cigarettes. The key independent variable is the dichotomous indicator of using a coupon and other price-related discounts during the most recent purchase (discounts). The covariates of other price minimisation strategies (OtherPMS) include four separate dichotomous variables, including the use of premium or generic brands in the past 30 days, purchase of latest cigarette by carton or by pack, purchase on Indian reservations during the previous

year and purchase through the internet during the previous year. These variables are included to control for the possibility of using overlapping strategies during the latest cigarette purchase. Daily smoking and time to first cigarette of the day (ciguse) are included in regression analysis as measures of smoking intensity and nicotine dependence so as to control for other price minimisation strategies that were not included in the survey, because heavy or more addicted smokers are more likely to use price minimisation strategies.[11 12 14 15] Daily smoking is an indicator of whether or not the respondent was a daily smoker (vs some-days smoker) at the time of the interview. Time to first cigarette after waking was a categorical variable of four (<5, 6–30, 31–60 and >60 min). Respondents' sociodemographic characteristics (demographics) and state dummy indicators (state) are also included to account for individual difference and state policy variation. Assessed respondents' sociodemographics is a vector which includes: gender (man or woman); age group (18–25, 26–44, 45–64 and 65+ years); race/ethnicity (non-Hispanic White, non-Hispanic Black, Hispanic, non-Hispanic Asian, non-Hispanic Native Hawaiian/Pacific Islander, non-Hispanic American Indian/Alaska Native and non-Hispanic 'Other'); education (less than high school, high school graduate or equivalent, some college and college degree or higher); marital status (married or cohabitate; widowed, divorced or separated and not currently in a relationship) and employment status (employed or unemployed).

Thus, the constant, $\beta_1$, presents the adjusted average per pack price before using any price minimisation strategies, and the coefficient, $\beta_2$ reflects the price reduction associated with price-related discounts directly from the industry. All analyses were performed using STATA (V.13). Post-stratification sampling weights were incorporated in all analyses to account for the complex survey design of the 2009–2010 NATS and non-response.

### RESULTS

Overall, among 11 766 adult current smokers, 38.4% identified Marlboro as the brand they used most often (figure 1), followed by Newport (15.1%) and Camel (8.7%). The percentage of users of other identified brand names were all less than 5%, respectively, ranging from Pall Mall (4.9%) to GPC (0.5%). The combined remaining 15.2% of smokers usually smoked cigarette brands (classified as other brands) that were not identified in the 2009–2010 NATS.

Table 1 presents the use of coupons or other price-related discounts among adult smokers by cigarette manufacturers. Specifically, 43.4% (4850) reported usually smoking cigarettes produced by PM, 23.9% (3274) usually smoked cigarettes from RJR and 15% (960) usually smoked Newport cigarettes (Lorillard; table 1). The remaining 17.7% (2682) smoked cigarettes from other companies, including Sonoma and USA Gold, which do not belong to the three major cigarette

**Figure 1** Brand preference among US adult smokers (2009–2010 NATS).

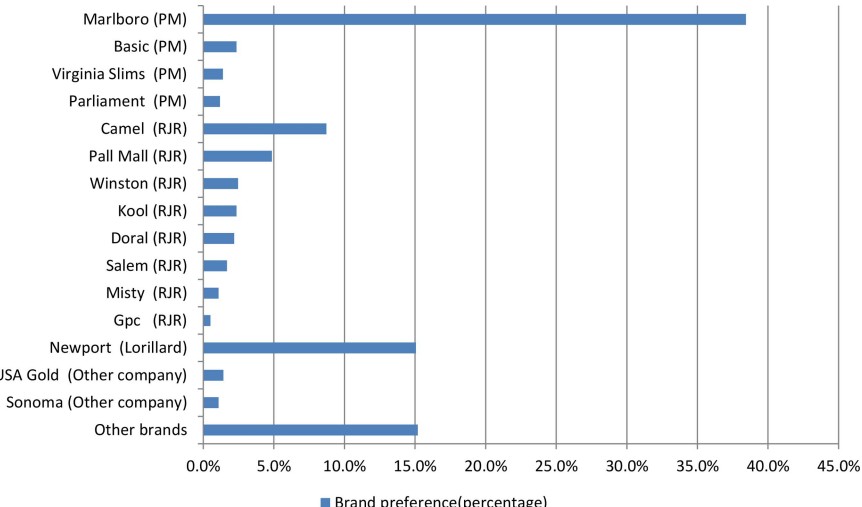

companies identified in the analysis. Approximately 24.4% of US adult smokers who smoked PM brands used coupons or other price-related discount offers from the company during their most recent cigarette purchase, compared with 21% of those who smoked RJR brands, 13.7% of those who smoked Newport (Lorillard) and 11.1% of those who smoked brands from other cigarette companies.

After adjusting for the use of multiple price minimisation strategies, respondents' demographic characteristics, smoking intensity and state policy variations, the average per pack prices paid for cigarettes from PM, RJR, Newport (Lorillard) and other companies were $5.06, $4.63, $4.75 and $3.94, respectively. The average price reductions of coupons or other discounts offered by PM or RJR were statistically significant and similar in magnitude, 29 cents per pack. That of Newport

(Lorillard) cigarettes was 24 cents per pack and was marginally significant.

Table 2 presents the use of coupons or other price-related discounts by specific brands (10 self-reported leading brands only) from the three leading cigarette manufacturing companies in the USA. Table 2 is ordered by PM brands first, followed by RJR brands and last by Newport, the only Lorillard brand listed. The prevalence of using coupons or other price-related discounts varied from 33.9% (Camel, RJR), to 25.6% (Marlboro, PM), 13.7% (Newport, Lorillard) and 10.5% the lowest (Salem, RJR). Thus, about one of three Camel smokers used these strategies during the last time they bought their cigarettes, compared with about one of four Marlboro smokers and about one of eight Newport smokers. The per cent price reductions due to use of coupons or discounts ranged from 1.1% (Basic)

**Table 1** The use of in-store coupons or other price-related discounts by major cigarette manufacturers*

| | PM | RJR | LORILLARD (Newport only) | Other companies† |
|---|---|---|---|---|
| Prevalence of brand use (%) | 43.4 | 23.9 | 15.0 | 17.7 |
| Prevalence of coupons or discounts used by smokers of that manufacture (%) | 24.4 | 21.0 | 13.7 | 11.1 |
| Price reduction per pack for smokers of that manufacture ($) | −0.29‡ | −0.29‡ | −0.24§ | −0.23 |
| Average price per pack for smokers of that manufacture ($) | 5.06 | 4.63 | 4.75 | 3.94 |
| Percentage of discount rendered to smokers of that manufacture (%) | 5.7 | 6.3 | 5.1 | 5.8 |
| N | 4850 | 3274 | 960 | 2682 |

Notes: N represents unweighted sample size. All estimates were obtained with post-stratification weights. In regressions, the dependent variable is price paid per pack, and the variable of interest is whether using coupons, rebates, buy-one-get-one-free, two-for-one or any other special promotions for cigarettes in the most recent purchase. Control variables include demographic characteristics (race, age, gender, education, marital status and employment status), state dummy variables, whether the respondent is a daily smoker, time to first cigarette since wake up, and all other price-minimisation behaviours (ie, purchase of generic brands, purchase of cartons, purchase on Indian reservation or purchase through the internet). Percentage of discount rendered is obtained by dividing average price per pack with price reduction associated with coupons and other price-related discounts.
*The self-reported use of coupons or other price-related discounts only reflects direct-to-consumer discounts from the industry. Therefore, industry's promotional allowances directly paid to retailers or wholesalers are not included.
†Users of Sonoma, USA Gold brands and other brand names which are not listed in the survey.
‡Statistically significant at 5% level.
§Statistically significant at 10% level.
PM, Philip Morris; RJR, R.J. Reynolds.

**Table 2** The use of in-store coupons or other price-related discounts by cigarette manufacturers and by top 10 leading US brands*

|  | Prevalence of coupon use (%) | % Of discount | Average price | Price after discount | Rank† |
|---|---|---|---|---|---|
| **PM** | | | | | |
| Basic | 22.2 | 1.1 | $4.41 | $4.36 | 3 |
| Marlboro | 25.6 | 5.6 | $5.00 | $4.72‡ | 2 |
| Virginia Slims | 10.6 | 3.4 | $6.49 | $6.27 | 9 |
| **RJR** | | | | | |
| Camel | 33.9 | 7.8 | $5.24 | $4.83‡ | 1 |
| Doral | 15.6 | 12.7 | $3.93 | $3.43‡ | 6 |
| Kool | 11.6 | 17.3 | $4.23 | $3.50‡ | 8 |
| Pall Mall | 15.9 | 1.3 | $3.19 | $3.15 | 5 |
| Salem | 10.5 | 15.7 | $5.11 | $4.31‡ | 10 |
| Winston | 17.1 | 3.7 | $4.07 | $3.92 | 4 |
| **Lorillard** | | | | | |
| Newport | 13.7 | 5.1 | $4.75 | $4.51§ | 7 |

N represents unweighted sample size. All estimates were obtained with post-stratification weights. In regressions, the dependent variable is price paid per pack, and the variable of interest is whether using coupons, rebates, buy-one-get-one-free, two-for-one or any other special promotions for cigarettes in the most recent purchase. Control variables include demographic characteristics (race, age, gender, education, marital status and employment status), state dummy variables, whether the respondent is a daily smoker, time to first cigarette since wake up and all other price-minimisation behaviours (ie, purchase of generic brands, purchase of cartons, purchase on Indian reservation or purchase through the internet). Percentage of discount rendered is obtained by dividing average price per pack with price reduction associated with coupons and other price-related discounts.
*The self-reported use of coupons or other price-related discounts only reflects direct-to-consumer discounts from the industry. Therefore, industry's promotional allowances directly paid to retailers or wholesalers are not included.
†Based on prevalence of coupon use.
‡Price reduction associated with coupons and other price-related discounts is statistically significant at 5% level.
§Statistically significant at 10% level.

to 17.3% (Kool). Among these 10 leading US cigarette brands, the average price paid per pack not using any price minimisation strategies ranged from paying $3.19 (Pall Mall, RJR) to $6.49 (Virginia Slims, PM). After using coupons or other price-related discounts, the average prices paid for a pack of cigarettes of the top three selling brands were 28 cents less for Marlboro ($4.72 instead of $5), 41 cents less for Camel ($4.83 instead of $5.24) and 24 cents less for Newport ($4.51 instead of $4.75). Also, those who smoked Salem and used coupons or other price-related discounts saved 80 cents the last time they purchased it. Finally, among these 10 leading US brands, users of Camel (RJR), Marlboro (PM) and Basic (PM) used price minimisation strategies the most.

## Discussion

Coupons or other price-related discounts from PM, RJR and Lorillard were used by 14–25% of their consumers and have provided price reductions for the smokers who used specific brands. Coupons or other price-related discounts from other companies did not result in statistically significant price per pack reductions for their consumers. Therefore, after controlling for the use of other price minimisation strategies and respondents' smoking intensity and nicotine addiction, the three leading cigarette companies provided price reductions for their products through coupons or other price-related discounts.

To put the range of price reductions associated with coupons or other price-related discounts into context, the cigarette federal tax was $1.01 per pack starting 1 April 2009, and the weighted average state cigarette excise tax rate was $1.17 per pack in 2010. These numbers imply that the coupon discounts from the three leading cigarette companies (about 24–29 cents) offset 23.8–28.7% of the price impacts from the federal tax or 11–13.3% of the price impacts from the federal and state excise taxes combined to the smokers of cigarettes produced by these manufacturers. As a result, these offers brought actual average prices down for users of specific brands, while the price reductions associated with these discounts were likely to increase the sales in these companies. For example, with a price elasticity of overall demand for cigarettes among adults at somewhere between −0.3 and −0.7,[22] ceteris paribus, these price discounts can be translated into 10.2–23.9 million packs of cigarette sales in 2010 for Marlboro, 3.0–6.6 million packs for Newport and 1.8–4.1 million packs for Camel.[21]

Although it is true that statistically significant or marginally statistically significant price reductions were observed for the three leading US cigarette companies, significant reductions are brand specific. The significant reductions were observed only for Marlboro (US leading brand), Camel (second US leading brand), Kool, Doral and Salem, while marginally for Newport (third US leading brand). Thus, PM, RJR and Lorillard have concentrated their efforts to provide price discounts mainly

to their best selling brands. This might be one of the reasons that a previous analysis failed to identify significant price reductions associated with promotional offers.[12]

This study has some limitations. First, the study design is cross-sectional. Therefore, the study findings may be specific only to the period October 2009–June 2010. However, the study covers the entire USA, and there is a variety of prices as a function of brand smoked and other factors. Second, most variables in the analysis are collected from recent purchases (ie, price paid, coupon and other price-related discounts and carton purchase are for the latest purchase; premium or generic brands are for the last 30 days), but others are collected with a time frame of 1 year (ie, purchase on Indian reservations, purchase through internet). However, when we excluded Indian reservation and internet purchases from the analysis, the adjusted average prices and price discounts associated with coupons did not change much. Third, because of an approval delay, only approximately 20% of respondents interviewed during the first 2 months of the survey were asked if purchases had been made on an Indian reservation anytime during the past year. In subsequent months, this question was asked for more than 90% of respondents. In the full sample, total missing responses for this question were 18.4% (3503). However, sensitivity analysis has shown that dropping these observations does not significantly affect the results.[16] As noted in the Methods section, we also excluded respondents who failed to report price paid for their latest purchase, the brand name they used most often in the past 30 days or some of their demographic characteristics. We compared smoking and social-demographic characteristics between individuals with incomplete information and individuals with complete information and have found little differences at the mean level. Fourth, as noted above, the 2009–2010 NATS does not collect information for all price minimisation strategies, including cigarette purchases from states with lower price. Although the cross-border purchase is an issue in tobacco control, the prevalence of this behaviour was quite low in the USA compared with other forms of price minimisation strategies. For example, data from the 2003 and 2006–2007 Tobacco Use Supplement to the Current Population Survey (TUS-CPS) suggest that about 5% of smokers made purchase across a state border,[23] while in the 2010–2011 TUS-CPS, approximately 3% of smokers purchased cigarettes from non-tribal land in lower-taxed and non-residential states (estimates not shown). In order to account for unmeasured price minimisation strategies, we controlled for smoking intensity and level of nicotine addiction in the analysis since the literature suggests that these are important risk factors of using any price minimisation strategies. Additionally, the paid prices were determined by using self-reported information from the smoker, which may be subject to recall bias. However, existing evidence indicates that the average of self-reported prices per pack in the 2009–2010 NATS was very consistent with the corresponding 2009 national average price reported in the Tax Burden on Tobacco (TBOT).[16] Another benefit of using self-reported price in this analysis is that we are able to control for the corresponding smoking intensity of each smoker, which is not available in market-scanned data but closely related to potential use of unmeasured price minimisation strategies. Finally, smokers' self-reported use of coupon or other types of price-related discounts in the 2009–2010 NATS only reflects direct-to-consumer discounts from the industry. As a result, the discount measure in the analysis does not include tobacco industry's promotional allowances directly paid to cigarette retailers or wholesalers, as these disaggregated promotional allowances by cigarette manufacturers or by brands have not been disclosed to consumers or to the public.

In addition to cigarette companies directly influencing cigarette retail prices by providing coupons or other price-related discounts, companies may also indirectly affect cigarette prices by offering discounts to retailers and by promoting cigars or pipe tobacco that can be used in roll-your-own cigarettes.[24] Although these indirect influences are critical in tobacco control and certainly warrant additional studies, the NATS survey did not collect such information thus they are not within the scope of this analysis.

Our results show that the three leading cigarette companies in the USA continue to offer price discounts to smokers of their brands, although these promotions appear to be concentrated among their top-selling cigarette brands. Cigarette companies can be strategic when offering price discounts. For example, existing literature suggests that young adults, women and heavy smokers are more frequently targeted for these promotions.[19] Other studies have shown that cigarette brands with low market share target young adults with the goal of encouraging brand switching, while major brands target older smokers to facilitate brand loyalty.[25] As pointed out earlier, smokers who use price-related discounts are less likely to make quit attempts or to successfully quit in the future. The length (duration) of smoking and the amount of cigarettes smoked per day on days the person smoked is strongly associated with a higher likelihood of developing and dying from a smoking-related disease, such as lung cancer, chronic obstructive pulmonary disease or heart attacks.[1] Therefore, even though price minimisation strategies may increase sales and profits for cigarette companies, these price discounts are likely preventing or delaying some smokers from permanent cessation. Policies that decrease price-minimisation strategies will benefit public health.

**Contributors** All the authors meet the authorship criteria: RSC participated in the conception and design of data; drafting the article or revising it critically for important intellectual content and final approval of the version to be published. XW participated in the conception and analysis and interpretation of data; revising it critically for important intellectual content and final approval of the version to be published. XX participated in the conception and

design of data; drafting the article or revising it critically for important intellectual content and final approval of the version to be published.

**Funding** This research received no specific grant from any funding agency in the public, commercial or not-for-profit sectors.

**Competing interests** None.

**Provenance and peer review** Not commissioned; externally peer reviewed.

**Data sharing statement** No additional data are available.

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
