## [Reviewer comments · BMJ Open]

Some articles will have been accepted based in part or entirely on reviews undertaken for other BMJ Group journals. These will be reproduced where possible.

ARTICLE DETAILS

TITLE (PROVISIONAL)	Can you refuse these discounts? An evaluation of the use and price discount impact of price-related promotions among U.S. adult smokers by cigarette manufacturer
AUTHORS	Caraballo, Ralph; Wang, Xu; Xu, Xin

VERSION 1 - REVIEW

REVIEWER	Kelvin Choi National Institute of Minority Health and Health Disparities
REVIEW RETURNED	26-Feb-2014

GENERAL COMMENTS	The authors used NATS 2009-2010 data to estimate the prevalence of use of discounts by smokers by manufacturer and brand, and the "effects" of these discounts. Overall, I am not sure about the novelty and the importance of this manuscript, given the authors already published another paper looking at strategy-specific cigarette cost-saving. I also have the following concerns: 1. "Effects" are not clearly defined in the abstract. I believe they investigated the amount of discount/cost-saving.2. While the authors only focused on coupons and in-store promotion, the ranking by tobacco manufacturer and brand can mislead the reader to believe that PM is not providing as much discount as RJR. They just have different discounting strategies. PM, in addition to direct-to-customer discounts, they also discount the price at the wholesale level, which is not reflected by study. That's why it is okay to look at effect of price discount on cigarette price at a whole, not by brand.3. The conceptualization of the study may not allow them to estimate the discount per brand. The ideal setup would be to have an observer go into a store and purchase all 15 brands on the same day, and record the sticker price (without discount) and retail price (with discounts), and noted the discount available. The issue with the current study is that smokers usually buy ONE brand of cigarette at ONE location (particularly when the study only limit to the last purchase). The authors tried to compensate this design limitation by statistical adjustment, which I don't think is doing the job. For example, even within a state, prices of cigarettes vary by neighborhood (without discount). Simply adjusting for state is not going to make the purchases comparable.4. A third of the smokers were excluded from the analysis. The authors only cited a study (with one sentence) to address this potential source of bias.
--

	5. The evidence related to effect of price-minimizing strategies and cessation behaviors should include two additional Minnesota cohort studies, published in Tobacco Control. 6. The authors also seem to have neglected the fact that use of coupons and promotion are subject to availability and also the starting price per pack of cigarettes, which also varies by brand. They did not reflect on this during their discussion. 7. It is unclear to me how the statistical analysis was done. I assumed the authors included price for a specific brand as outcome, and examined the associated between use of coupon/promotion and price. So each brand/manufacturer was modeled separately. But this was not mentioned in the manuscript. 8. Given the text and the distribution, Figure 1 is not necessary. 9. I am not sure the average price is the right point of comparison. The average price adjusting for price-minimizing strategies means it is average across those who did and did not use price-minimizing strategies. A better comparison point should be estimated price when all price-minimizing strategy variables are set to zero.
--	---

REVIEWER	Laura Cornelsen London School of Hygiene & Tropical Medicine, United Kingdom
REVIEW RETURNED	14-Mar-2014

GENERAL COMMENTS	 - The objective is not clear because it says the aim is to "evaluated the uses and effects of these price-related discounts by...." I think it should state clearly that the work is looking at the impact of the discounts on the price paid by smokers. The study does not go beyond to evaluate the possible effect of this on consumption or health. - Output from the regression analysis should be presented (if not in the main text then perhaps as an online appendix?) as it would be interesting to see the role/significance of other variables -> e.g. what is the difference in the price paid across the demographic variables you have included. It would also allow to assess the overall fit and significance of the model. - Page 11 rows 9-15: I think this conclusion is too strong. All you can really say is that coupon discounts offset 24-29% of the federal tax not public health impacts. The response from price reduction to public health impact is not 1:1. On that note, you could do simple calculations to show how much ceteris paribus consumption is likely to increase due to discounts as it is well established that the price elasticity of demand for tobacco is around 0.2 to 0.4 in high-income countries. - You have reported that NATS is nationally representative but it is not clear whether the sample you use remains so as it is considerably smaller. You report using NATS national weights presumably to make results representative at national level. I'm not familiar with NATS but stratified samples usually have sampling weights created according to the stratification scheme. You have drawn your sample based on a specific question which doesn't necessarily follow the stratification. Even if it did, using the general sampling weights without adjustment, which is not mentioned, to the smaller sample size would lead to underestimation at population level. I would like to see more explanation on the use of weights and
--

	the representativeness of the sample.  - The overall conclusion of the study could be stronger, directly saying that tobacco control policy should target to eliminate the discount possibilities. It's clear from demand studies that reduction in price (which discounts cause as you show) is associated with higher consumption of tobacco. - Overall I think it is a useful study and deserves publication as there is not a great deal of insight available on prices paid at individual level, particularly emphasizing the use of discount strategies.
--	---

VERSION 1 – AUTHOR RESPONSE

The authors used NATS 2009-2010 data to estimate the prevalence of use of discounts by smokers by manufacturer and brand, and the "effects" of these discounts. Overall, I am not sure about the novelty and the importance of this manuscript, given the authors already published another paper looking at strategy-specific cigarette cost-saving. I also have the following concerns:

1. "Effects" are not clearly defined in the abstract. I believe they investigated the amount of discount/cost-saving.

Response: Thank you, Dr. Kelvin Choi. We have made edits as suggested.

2. While the authors only focused on coupons and in-store promotion, the ranking by tobacco manufacturer and brand can mislead the reader to believe that PM is not providing as much discount as RJR. They just have different discounting strategies. PM, in addition to direct-to-customer discounts, they also discount the price at the wholesale level, which is not reflected by study. That's why it is okay to look at effect of price discount on cigarette price at a whole, not by brand.

Response: We agree with the reviewer that smokers' self-reported use of coupon or other types of price-related discounts in the 2009-2010 National Adult Tobacco survey (NATS) only reflects direct-to-consumer discounts from the industry. As a result, the discount measure in the analysis does not include tobacco industry's promotional allowances directly paid to cigarette retailers or wholesalers, as these disaggregated promotional allowances by cigarette manufacture or by brand have not been disclosed to consumers or to the public. To avoid potential confusion, we include additional table notes under Tables 1 and 2 to clarify the situation.

However, with all due respect, although the self-reported discount/coupon use does not reveal the full picture of industry's marketing strategy, we believe price discount information by cigarette manufacture or by brand reported in Tables 1 and 2, including both the prevalence of use and the magnitude of discount are very important findings of the analysis, because

- (1) Tobacco industry is strategic when offering direct-to-consumer price promotions. Cigarette brands with low market share target younger individuals with the price promotions of free packs of cigarettes and coupons, with the goal of encouraging brand switching.[1] Major cigarette brands, on the other hand, target the price promotion of free packs of cigarettes to older smokers to encourage them to continue purchasing the major brand and to discourage cigarette quitting[1].
- (2) Unlike promotional allowances to cigarette retailers or wholesalers, direct-to-consumer discounts allow cigarette manufactures to target price promotions to specific populations, including individuals with a higher propensity to quit smoking.[2] Therefore, this strategy is of particular interest in tobacco control regulatory efforts.
- (3) Above all, existing literature provides very limited insights to direct-to-consumer discounts by manufacture or brand.

3. The conceptualization of the study may not allow them to estimate the discount per brand. The ideal setup would be to have an observer go into a store and purchase all 15 brands on the same day, and record the sticker price (without discount) and retail price (with discounts), and noted the discount available. The issue with the current study is that smokers usually buy ONE brand of

cigarette at ONE location (particularly when the study only limit to the last purchase). The authors tried to compensate this design limitation by statistical adjustment, which I don't think is doing the job. For example, even within a state, prices of cigarettes vary by neighborhood (without discount). Simply adjusting for state is not going to make the purchases comparable.

Response: This is a very stimulating comment. More research is certainly warranted to explore variations in cigarette prices by neighborhood. But to clarify, in our analysis, in addition to state fixed effect, we include covariates of smokers' demographic characteristics, such as race, age, gender, education, marital status, and employment status, as well as nicotine addiction measures, such as whether the respondent is a daily smoker and time to first cigarette since wake up. These demographic measures at individual level all together provide a reasonable control for respondents' neighborhood characteristics.

We also agree with the reviewer that, in an alternative situation, someone could perform experiments by sending pseudo smokers to purchase all cigarette brands in local retail stores and online to collect the discount information. However, we are not sure if this experiment can provide enough insights for the reality of the situation.

Since one day experiment can hardly reveal the overall picture of industry's price-promotion strategies throughout the year, frequency of the experiments as well as the costs of the experiments certainly need to be considered. The scale of the experiment can be another challenge. The 2009-2010 NATS is a nationally representative survey of tobacco consumption among U.S. adults. Building a national marketing surveillance system based on the experiments can be very costly and is very less likely to be a cost-effective strategy. Besides, without actual local sales information by manufacture or by brand, the discount information alone can not reveal the market share, brand interests, or the actual use of these price-promotion strategies among smokers in the local market.

4. A third of the smokers were excluded from the analysis. The authors only cited a study (with one sentence) to address this potential source of bias.

Response: We agree with the reviewer that the sample size of complete cases used in this analysis was smaller than the original NATS sample. This is mainly because of the missing values in household income, and the missing in the Indian reservation question due to an approval delay in the survey process. We acknowledge that issue in the limitation section.

In the online appendix of the cited article (available at <http://www.sciencedirect.com/science/article/pii/S0749379713001098>), an early study based on the same sample, we performed a couple of different sensitivity analyses. The sensitivity analyses that compared dropping the observations rather than treating observations with missing data on Indian reservation question as non-Indian reservation purchases showed no bias. The national price for those not practicing a price-minimization strategy would only be 0.7% less if these observations were excluded instead of being treated as non-Indian reservation purchases. In state specific analysis, only two states, South Dakota and Tennessee, showed a price difference for this measure of greater than 5% compared to reported prices (-5.1% and 5.9%, respectively).

In another newly published article using the same data,[3] we also examined the impact of missing data in independent variables, including both Indian reservation question and household income among others, in the sensitivity analysis by re-estimating the model with categorical variables for any missing data. The price reductions associated with price minimization strategies differed by at most 2 cents and none of these differences were statistically significant from the original estimates, which were based on complete cases.

The results of these sensitivity analyses indicate that little difference in the size of price reductions for individuals with and without complete information.

For comparison purpose for this analysis, we also provided a table to compare social-demographic characteristics between complete cases and samples with any missing values. These results indicate that although little differences exist between the complete cases and samples with missing values, these differences are pretty small compared to the mean values.

Response Table 1: The comparison of social-demographic characteristics between complete cases and samples with any missing values

	Complete cases	Sample with missing value	Mean difference
Female	0.557 (0.497)	0.561 (0.496)	-0.004 (0.009)
Non-Hispanic black	0.100 (0.300)	0.0839 (0.277)	0.016** (0.005)
Hispanic	0.0429 (0.203)	0.0456 (0.209)	-0.003 (0.004)
Non-Hispanic Asian	0.00875 (0.0932)	0.00810 (0.0897)	0.001 (0.002)
Native Hawaiian or Pacific Islander	0.00629 (0.0791)	0.00547 (0.0738)	0.001 (0.001)
American Indian or Alaska Native	0.0354 (0.185)	0.0348 (0.183)	0.001 (0.003)
Other races	0.0264 (0.160)	0.0256 (0.158)	0.001 (0.003)
Age 18 to 25	0.0862 (0.281)	0.0739 (0.262)	0.012* (0.005)
Age 45 to 64	0.473 (0.499)	0.484 (0.500)	-0.011 (0.009)
Age 65 or older	0.133 (0.340)	0.168 (0.374)	-0.035*** (0.006)
Less than high school	0.134 (0.341)	0.133 (0.339)	0.002 (0.006)
Some college or associate degree	0.352 (0.478)	0.370 (0.483)	-0.018* (0.008)
Bachelor degree or higher	0.188 (0.391)	0.208 (0.406)	-0.020** (0.007)
Widowed divorced or separated	0.305 (0.460)	0.310 (0.463)	-0.006 (0.008)
Never married	0.209 (0.407)	0.198 (0.399)	0.011 (0.007)
Currently unemployed	0.464 (0.499)	0.490 (0.500)	-0.026** (0.009)
Observations	11,766	4,776	16,542

Notes: Estimates are un-weighted mean; standard deviations are in parentheses; Estimates in column 3 are mean difference between non-missing sample and missing sample. * p<0.05 ** p<0.01 *** p<0.001. Missing sample includes respondents who failed to report a brand name (n=523), who smoked Forsyth (n=4), those who failed to provide information on price paid for their latest purchase (n=978), the use of price-minimization strategies (n=2,794), demographic characteristics (age, race/ethnicity, gender, education, marital status, or employment status), or time to first cigarette since wake up (n=477).

5. The evidence related to effect of price-minimizing strategies and cessation behaviors should include two additional Minnesota cohort studies, published in Tobacco Control.

Response: We have cited the two articles as the reviewer suggested.

6. The authors also seem to have neglected the fact that use of coupons and promotion are subject to availability and also the starting price per pack of cigarettes, which also varies by brand. They did not reflect on this during their discussion.

Response: We have made edits as the reviewer suggested. A few statements were included in the last paragraph of the discussion section to note the availability of price-related discounts by brand.

7. It is unclear to me how the statistical analysis was done. I assumed the authors included price for a specific brand as outcome, and examined the association between use of coupon/promotion and price. So each brand/manufacture was modeled separately. But this was not mentioned in the manuscript.

Response: We have made edits in the method section to clarify the methodology. Specifically, we included the regression specification of the statistical analysis. We also clarified that the regression analysis has been done separately by brand/manufacture.

8. Given the text and the distribution, Figure 1 is not necessary.

Response: We think Figure 1, brand preference among U.S. adult smokers, is important to include in the manuscript, since the first paragraph in the result section does not provide a full market share description of all the 15 cigarette brands included in the analysis due to the word limitation.

More importantly, the figure together with estimates reported in Table 2 offers an opportunity for the audience to independently assess the association between price discounts and brand preference among U.S. adult smokers by themselves.

9. I am not sure the average price is the right point of comparison. The average price adjusting for price-minimizing strategies means it is average across those who did and did not use price-minimizing strategies. A better comparison point should be estimated price when all price-minimizing strategy variables are set to zero.

Response: We agree with the reviewer that the average price is not the appropriate base for the comparison. However, to clarify, we do not compare price discounts by manufacture or by brand with the corresponding average price in the analysis. In contrast, we use the *adjusted* average per pack price paid by smokers, *meaning* the average price after controlling for all price-minimizing strategies.

As we illustrated in the method section, to assess the independent price reduction associated with coupons and other price-related discounts, regression analysis with the following specification has been conducted for each cigarette company or brand,

$$\text{Per pack price paid} = \beta_1 + \beta_2 \text{discounts} + \beta_3 \text{OtherPMS} + \beta_4 \text{Ciguse} + \beta_5 \text{Demographics} + \beta_6 \text{state}$$

We note that the estimated coefficient, β_2 in the specification, reflects the independent price reduction associated with price-related discounts by manufacture or by brand (discounts). The estimated coefficient β_3 denotes the price reductions associated with other price minimization strategies (otherPMS), including the use of generic brands in the past 30 days, purchase of latest cigarette by carton or by pack, purchase on Indian reservations during the previous year, and purchase through the Internet during the previous year. These variables are included to control for the potential use of overlapping strategies, for example, use coupon and purchase by carton in the latest cigarette purchase.

Cigarette use variables (ciguse), including daily smoking and time to first cigarette of the day are included as measures of smoking intensity and nicotine dependence so as to control for other price minimization strategies that are not included in the 2009-2010 NATS, because heavy or more addicted smokers are more likely to use price minimization strategies.

As a result, the estimated constant, β_1 , presents the *adjusted* average per pack price paid *before using any price minimization strategies* and it (β_1) is consistent with the reviewer's suggestion, obtaining "estimated price when all price-minimizing strategy variables are set to zero."

Reviewer Name Laura Cornelsen

Institution and Country London School of Hygiene & Tropical Medicine, United Kingdom
Please state any competing interests or state 'None declared': None declared

- The objective is not clear because it says the aim is to "evaluated the uses and effects of these price-related discounts by...." I think it should state clearly that the work is looking at the impact of the discounts on the price paid by smokers. The study does not go beyond to evaluate the possible effect of this on consumption or health.

Response: Thank you, Dr. Laura Cornelsen for your comments. To clarify the objective of the analysis, we made edits as the reviewer suggested.

- Output from the regression analysis should be presented (if not in the main text then perhaps as an online appendix?) as it would be interesting to see the role/significance of other variables -> e.g. what is the difference in the price paid across the demographic variables you have included. It would also allow to assess the overall fit and significance of the model.

Response: We included the full output tables at the end of this response letter. We can also include those tables in an online appendix if the journal allows.

- Page 11 rows 9-15: I think this conclusion is too strong. All you can really say is that coupon discounts offset 24-29% of the federal tax not public health impacts. The response from price reduction to public health impact is not 1:1. On that note, you could do simple calculations to show how much ceteris paribus consumption is likely to increase due to discounts as it is well established that the price elasticity of demand for tobacco is around 0.2 to 0.4 in high-income countries.

Response: We revised the conclusion as the reviewer suggested. We rephrase it to the price impacts, instead of public health impacts. We acknowledge and agree with the reviewer that the response from price reduction to public health impact may not be 1:1.

As the reviewer suggested, we also included additional conclusions on the potential impacts of price discounts on cigarette sales in the conclusion section.

- You have reported that NATS is nationally representative but it is not clear whether the sample you use remains so as it is considerably smaller. You report using NATS national weights presumably to make results representative at national level. I'm not familiar with NATS but stratified samples usually have sampling weights created according to the stratification scheme. You have drawn your sample based on a specific question which doesn't necessarily follow the stratification. Even if it did, using the general sampling weights without adjustment, which is not mentioned, to the smaller sample size would lead to underestimation at population level. I would like to see more explanation on the use of weights and the representativeness of the sample.

Response: We revised the statement on the poststratified sampling weight in the method section. The NATS poststratified sampling weights incorporated in all analyses were intended to account for the complex survey design of the 2009-2010 NATS and nonresponse of the survey.

To be more specific about the 2009–2010 NATS sampling and sampling weights:

It is a stratified, national, dual-frame telephone survey. The NATS target population was noninstitutionalized adults aged 18 years or older residing in the 50 US states and the District of Columbia. The sample was designed to be representative at both national and state levels. Each state was divided into at least three strata: listed landline, unlisted landline, and cell phone. The listed stratum consisted of landline telephone numbers in residential directories or in other source

databases, whereas the unlisted stratum consisted of landline telephone numbers not listed as a residential number in any source database. Some states also had additional landline strata based on counties or county-equivalents. For the landline component, each state was allocated an equal target sample size (n=1,863) to ensure adequate precision for state-level estimates. For the cell phone component, each state was allocated a sample size in proportion to its population. Four states independently added to their samples: Louisiana, New Jersey, North Dakota, and Oklahoma.

Respondent selection varied by phone type. For landline numbers, one adult was randomly selected from each eligible household. Respondents who used only cell phones were selected through screening of a sample of cell phone numbers. In total, 118,581 interviews were completed (n=110,634 landline, n=7,947 cell phone) during October 2009–June 2010. The national Council of American Survey and Research Organizations (CASRO) response rate was 37.6% (landline=40.4%, cell phone=24.9%); the national cooperation rate was 62.3% (landline=61.9%, cell phone=68.7%).

The landline data were first weighted by the probability of selection of the telephone number, the probability of selecting the respondent, and a nonresponse adjustment. The cell phone data were initially weighted only by the probability of selection of the telephone number. Next, with the use of a raking procedure, the data were poststratified by state to the joint estimated distributions of age, gender, and phone type (cell-phone-only users and all others). Additional information about the NATS survey is available at http://www.cdc.gov/tobacco/data_statistics/surveys/nats/pdfs/methodology-report.pdf.

We agree with the reviewer that the sample size of complete cases used in this analysis was smaller than the original NATS sample. This is mainly because of the missing values in household income, self-reported per pack price paid, as well as the missing in the Indian reservation question due to an approval delay in the survey process.

In the online appendix of the cited article (available at <http://www.sciencedirect.com/science/article/pii/S0749379713001098>), an early study based on the same sample, we performed sensitivity analyses. The sensitivity analyses that compared dropping the observations rather than treating observations with missing data on Indian reservation question as non-Indian reservation purchases showed no bias. The national price for those not practicing a price-minimization strategy would only be 0.7% less if these observations were excluded instead of being treated as non-Indian reservation purchases. In state specific analysis, only two states, South Dakota and Tennessee, showed a price difference for this measure of greater than 5% compared to reported prices (-5.1% and 5.9%, respectively).

In another newly published article using the same data,[3] we also examined the impact of missing data in independent variables, including both Indian reservation question and household income among others, in the sensitivity analysis by re-estimating the model with categorical variables for any missing data. The price reductions associated with price minimization strategies differed by at most 2 cents and none of these differences were statistically significant from the original estimates, which were based on complete cases.

The results of these sensitivity analyses indicate that little difference in the size of price reductions for individuals with and without complete information.

For the comparison purpose of this analysis, we provided a table to compare social-demographic characteristics between complete cases and samples with any missing values. These results indicate that although little differences exist between the complete cases and samples with missing values, these differences are pretty small compared to the mean values.

Response Table 1: The comparison of social-demographic characteristics between complete cases and samples with any missing values

	Complete cases	Sample with missing value	Mean difference
Female	0.557 (0.497)	0.561 (0.496)	-0.004 (0.009)
Non-Hispanic black	0.100	0.0839	0.016**

	(0.300)	(0.277)	(0.005)
Hispanic	0.0429	0.0456	-0.003
	(0.203)	(0.209)	(0.004)
Non-Hispanic Asian	0.00875	0.00810	0.001
	(0.0932)	(0.0897)	(0.002)
Native Hawaiian or Pacific Islander	0.00629	0.00547	0.001
	(0.0791)	(0.0738)	(0.001)
American Indian or Alaska Native	0.0354	0.0348	0.001
	(0.185)	(0.183)	(0.003)
Other races	0.0264	0.0256	0.001
	(0.160)	(0.158)	(0.003)
Age 18 to 25	0.0862	0.0739	0.012*
	(0.281)	(0.262)	(0.005)
Age 45 to 64	0.473	0.484	-0.011
	(0.499)	(0.500)	(0.009)
Age 65 or older	0.133	0.168	-0.035***
	(0.340)	(0.374)	(0.006)
Less than high school	0.134	0.133	0.002
	(0.341)	(0.339)	(0.006)
Some college or associate degree	0.352	0.370	-0.018*
	(0.478)	(0.483)	(0.008)
Bachelor degree or higher	0.188	0.208	-0.020**
	(0.391)	(0.406)	(0.007)
Widowed divorced or separated	0.305	0.310	-0.006
	(0.460)	(0.463)	(0.008)
Never married	0.209	0.198	0.011
	(0.407)	(0.399)	(0.007)
Currently unemployed	0.464	0.490	-0.026**
	(0.499)	(0.500)	(0.009)
Observations	11,766	4,776	16,542

Notes: Estimates are un-weighted mean; standard deviations are in parentheses; Estimates in column 3 are mean difference between non-missing sample and missing sample. * $p < 0.05$ ** $p < 0.01$ *** $p < 0.001$. Missing sample includes respondents who failed to report a brand name ($n=523$), who smoked Forsyth ($n=4$), those who failed to provide information on price paid for their latest purchase ($n=978$), the use of price-minimization strategies ($n=2,794$), demographic characteristics (age, race/ethnicity, gender, education, marital status, or employment status), or time to first cigarette since wake up ($n=477$).

- The overall conclusion of the study could be stronger, directly saying that tobacco control policy should target to eliminate the discount possibilities. It's clear from demand studies that reduction in price (which discounts cause as you show) is associated with higher consumption of tobacco.

Response: Because all authors of this manuscript are federal government employees, such policy recommendation may not be appropriate.

- Overall I think it is a useful study and deserves publication as there is not a great deal of insight available on prices paid at individual level, particularly emphasizing the use of discount strategies.

VERSION 2 – REVIEW

REVIEWER	Kelvin Choi National Institute on Minority Health and Health Disparities, USA
REVIEW RETURNED	25-Apr-2014

GENERAL COMMENTS	Thanks for addressing my concerns in the revised manuscript. I think the manuscript has improved. A few additional suggestions: 1. I appreciate the footnotes under Table 1 and 2 on the limitation that the data do not reflect price-discounting strategies other than direct-to-consumer discounts. I would recommend expanding the section in the discussion a little more to address the issue that the ranking by brand and manufacturer does not reflect the ranking of total effort by manufacturer to manipulate price. The current footnotes do not clarify this point.2. The authors said that they address my concern about availability of price-related discounts by brand in the last paragraph of the discussion section, which I have a hard time finding those edits.3. The equation in the method section may or may not be helpful. While it may be helpful in pointing out B1, B3 makes it sound like uses of all other PMS are combined into a single beta, which I don't believe that was the case. Same for demographics.
--

REVIEWER	Laura Cornelsen London School of Hygiene and Tropical Medicine
REVIEW RETURNED	15-Apr-2014

- The reviewer completed the checklist but made no further comments.

VERSION 2 – AUTHOR RESPONSE

Reviewer Name Kelvin Choi

Institution and Country National Institute on Minority Health and Health Disparities, USA

Please state any competing interests or state 'None declared': None declared.

1. I appreciate the footnotes under Table 1 and 2 on the limitation that the data do not reflect price-discounting strategies other than direct-to-consumer discounts. I would recommend expanding the section in the discussion a little more to address the issue that the ranking by brand and manufacturer does not reflect the ranking of total effort by manufacturer to manipulate price. The current footnotes do not clarify this point.

Authors' response: The reviewer's suggestion is well taken. We have added a few statements at the end of the limitation section.

2. The authors said that they address my concern about availability of price-related discounts by brand in the last paragraph of the discussion section, which I have a hard time finding those edits.

Authors' response: We are sorry that the reviewer missed our highlighted edits in the discussion section. In the last paragraph of the discussion section, the newly added statements related to the availability of price-related discounts are "Cigarette companies can be strategic when offering price

discounts. For example, existing literature suggests that young adults, females, and heavy smokers are more frequently targeted for these promotions.¹⁹ Other study have shown that cigarette brands with low market share target young adults with the goal of encouraging brand switching, while major brands target older smokers to facilitate brand loyalty.²⁵

3. The equation in the method section may or may not be helpful. While it may be helpful in pointing out B1, B3 makes it sound like uses of all other PMS are combined into a single beta, which I don't believe that was the case. Same for demographics.

Authors' response: We have added additional description in the section of statistical analysis to clarify the variables we used for price-minimization strategies and demographic characteristics.